# Not Only Hydrogen Bonds: Other Noncovalent Interactions

**Ibon Alkorta** [1,*] **, José Elguero** [1,*] **and Antonio Frontera** [2,*]

[1] Instituto de Química Médica, CSIC, Juan de la Cierva, 3, E-28006 Madrid, Spain
[2] Departament de Química, Universitat de les Illes Balears, Crta. de Valldemossa km 7.5, 07122 Palma de Mallorca, Spain
[*] Correspondence: ibon@iqm.csic.es (I.A.); iqmbe17@iqm.csic.es (J.E.); toni.frontera@uib.es (A.F.)

**Abstract:** In this review, we provide a consistent description of noncovalent interactions, covering most groups of the Periodic Table. Different types of bonds are discussed using their trivial names. Moreover, the new name "Spodium bonds" is proposed for group 12 since noncovalent interactions involving this group of elements as electron acceptors have not yet been named. Excluding hydrogen bonds, the following noncovalent interactions will be discussed: alkali, alkaline earth, regium, spodium, triel, tetrel, pnictogen, chalcogen, halogen, and aerogen, which almost covers the Periodic Table entirely. Other interactions, such as orthogonal interactions and π-π stacking, will also be considered. Research and applications of σ-hole and π-hole interactions involving the p-block element is growing exponentially. The important applications include supramolecular chemistry, crystal engineering, catalysis, enzymatic chemistry molecular machines, membrane ion transport, etc. Despite the fact that this review is not intended to be comprehensive, a number of representative works for each type of interaction is provided. The possibility of modeling the dissociation energies of the complexes using different models (HSAB, ECW, Alkorta-Legon) was analyzed. Finally, the extension of Cahn-Ingold-Prelog priority rules to noncovalent is proposed.

**Keywords:** noncovalent interactions; Lewis acids; Lewis bases; spodium bonds; σ/π-hole interactions

## 1. Introduction

The aim of this review is to present an original, systematic and prospective view of all noncovalent interactions (NCI). There are several books treating different aspects of NCIs [1–4] but none offers a unified view of the subject, for instance the term Lewis acid/Lewis base does only appear in the most recent one [3]. See on this topic a recent conference paper entitled "Some interesting features of the rich chemistry around electron-deficient systems" [5].

We excluded hydrogen bonds from this survey on NCIs because they are well known and because the bibliography covering HBs is more extensive than the sum of the references on the other NCIs [6–11]. We also excluded anions and cations limiting this review to neutral molecules.

In the modified IUPAC periodic table of the elements reported in Figure 1, we noted in black all the NCIs reported up to now and in blue these not yet discussed. A similar representation was used by Caminati et al. for the front page of their publication [12]. They called the bonds of the groups MB (2), IB (13), TB (14), NB (15), CB (16), and XB (17) following previous authors.

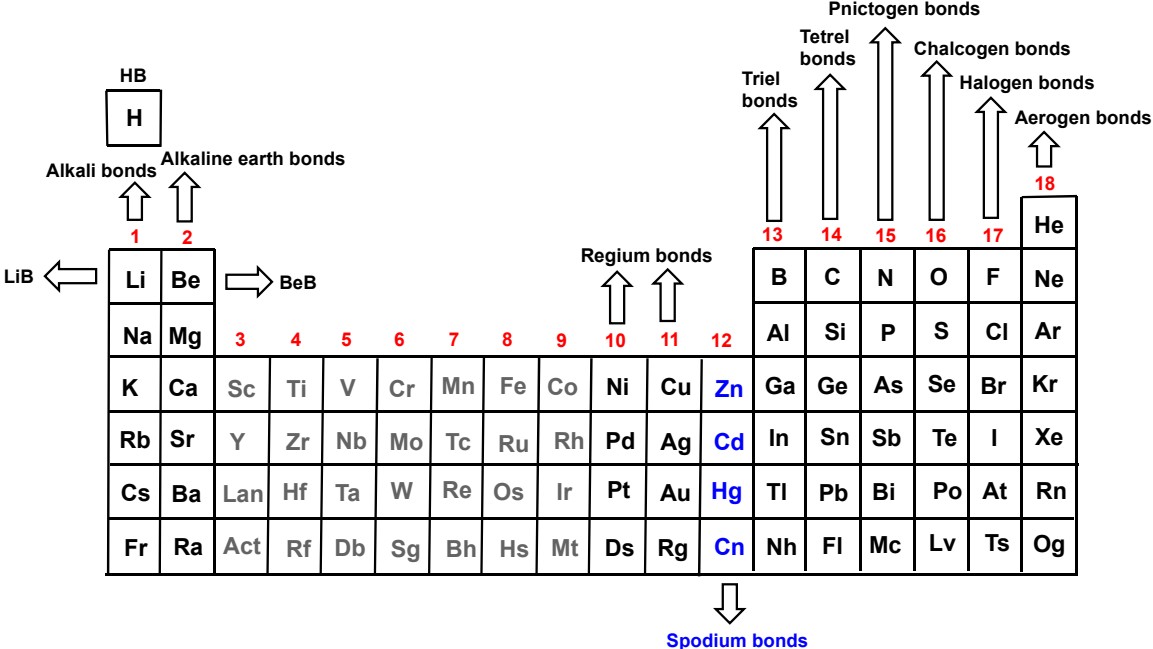

**Figure 1.** The different noncovalent bonds formed by elements of the Periodic Table. In black are accepted names, and in blue are the proposed new names. Groups 3 to 9 (in grey) are not included in this review.

Usually, the bond is associated with the Lewis acidity of a group, this is the case with groups 11, 13, 14, 15, 16, 17, and 18. For groups 1 and 2, besides HBs, the bond is associated to an element, lithium, sodium and beryllium. We propose to call these bonds Alkali Bonds and Alkaline Earth Bonds (we used this name very recently) [13]. Although Regium Bonds were used for group 11, we propose to use it for both 10 and 11 groups. In grey are the atoms corresponding to groups 3 to 9 that we will not discuss, not that they were unable to form NCIs, but in order not to stretch too much this mini review.

Concerning the rows, we should indicate that Li, Be, B, and C derivatives as Lewis acids have been more studied than Na, Mg, Al, and Si. On the other hand, P, S, and Cl are better representatives of their kind of NCIs than N, O, and F. This observation is related to size and to the softness of the Lewis acid atom that interacts with the Lewis base [14]

Gilbert N. Lewis published his interpretation of acid/base behavior in 1923 [15]; according to him any species with a reactive vacant orbital or available lowest unoccupied molecular orbital is classified as a "Lewis acid" [14,16].

A Lewis base (LB) is associated with a region of the space where there is an excess of negative charge (electron density) in the proximity of an atom or several atoms of a molecule. This happens in anions and in some neutral molecules, such as lone pairs (LP: carbenes, amines, phosphines, *N*-oxides, . . . ), multiple bonds (olefins, acetylenes, benzenes, and other aromatic molecules, . . . ), single bonds (alkanes, dihydrogen, . . . ), radicals, metals (rare), . . .

A Lewis acid (LA) is associated with a region of the space where there is an excess of positive charge (a deficit of negative charge, electron deficiency) in the proximity of an atom or several atoms of a molecule. This happens in cations, σ- and π-holes, metals (frequent), . . . The concepts of σ-hole and π-hole were introduced by Politzer et al. [17–19] to describe regions of positive potential along the vector of a covalent bond (σ-hole) or perpendicular to an atom of molecular framework (π-hole).

Some atoms have simultaneously (but in different parts of the space) LB and LA zones due to their anisotropic distribution of electron density. The same happens for molecules, but in this case, they correspond to different parts of the molecule. Note that some Lewis acids when interacting with stronger Lewis acids can behave as Lewis bases [20].

When an LB and an LA containing atoms or molecules are free to interact (i.e., non restrained by some geometrical hindrance), they form complexes being their minima or transition states of different order.

The information on NCIs is mostly based on from crystal structures, microwave (MW) spectroscopy and theoretical calculations; consequently, they are related to gas-phase and solid state. Since chemistry is mainly done in solution there is a consistency problem.

Another aspect that is common to all NCIs is cooperativity. The natural evolution of theoretical studies has been moving from dimer complexes to trimers and longer complexes in search of cooperativity, both augmentative and diminutive, present in crystal structures.

Definition: Noncovalent interactions are complexes formed by two or several LBs and LAs. It is the LA that gives the name to the interaction. Dative bonds are included in this definition.

Why were the complexes not named according to the LB? Historically, because all NCI derive from HBs, i.e., where the H-bond donor is the Lewis acid. More fundamentally, it is because it is not possible to define families of NCIs based on LB. For instance, all anions are LBs, and anions can be found all over the Periodic Table. A classification of LBs is given in Figure 2.

| Anions | Neutral molecules | | | |
|---|---|---|---|---|
| | Atoms | Bonds | Rings | Groups |
| $H^-$<br>$F^-$<br>$CN^-$<br><br>⬠⁻ | He | H≡≡≡H<br><br>H—H | ⬡ | ⬡N:<br><br>$BrCH_3$ |

**Figure 2.** Lewis bases involved in noncovalent interactions.

The proposed definition allows naming immediately the famous $H_3N:BH_3$ complex [21]; since $BH_3$ is the LA, this is an example of triel bond. The recent controversy Zhou-Frenking/Landis-Weinhold on the $Ca(CO)_8$ complex [22–24] leads us to propose the classify them as alkaline earth bonds, the CO being the Lewis bases.

In a recent paper, it is written: "It is well known that alkynes act as π-acids in the formation of complexes with metals" [25]. If this were correct, then the bond should be a tetrel one; on the other hand, if the alkyne was the base and the metal (in this case Au) the Lewis acid [14], the bond would be a regium bond.

This review does not try to discuss the nature of the bonds [26] we classified as NCIs. This is still a subject not settled [27]. For instance, Mo et al., using the block-localized wave function (BLW), analyzed the halogen bond [28], concluding that it is a charge transfer (CT) interaction, i.e., an intermolecular hyperconjugation consistent with Mulliken proposal [29]. The same authors used the BLW methodology to analyze hydrogen, halogen, chalcogen, and pnictogen bonds, stressing the magnitude of covalency, directionality, and σ-hole concept [30]. A review by Jin et al. [31] compared the σ-hole and π-hole bonds based on halogen bonds. Grabowski et al. [32] discussed halogen, chalcogen, pnictogen, and tetrel bonds as LA-LB complexes.

## 2. Alkali Bonds

The oldest of NCIs (not including HBs) are the *Halogen Bonds* that, although not named like this, were reported in 1948–1950 by Benasi, Hildebrand, and Mulliken [29,33]. *Lithium Bonds* were introduced by three great chemists: Kollman, Liebman, and Allen in 1970 [34]. We contributed with a

paper [35] to this field, where we studied F–Li$\cdots$N, H–Li$\cdots$N and H$_3$C–Li$\cdots$N lithium bonds. The set of nitrogen Lewis bases consists of two that are sp hybridized (N$_2$ and HCN); five sp$^2$-hybridized bases, four of which are aromatic (1,3,5-triazine, 1,2,3-triazine, pyrazine, and pyridine), one nonaromatic (HN=CH$_2$); and three sp$^3$-hybridized bases (NH$_3$, NH$_2$CH$_3$, and aziridine).

There have been two theoretical papers reporting *Sodium bonds* [36,37] but, so far, none reporting *Potassium bonds*. For consistency reasons, we propose to call all of them *Alkali bonds*. The paper on sodium bonds reported cooperativity between halogen and sodium bonds in NCX$\cdots$NCNa$\cdots$NCY complexes, where Y = F, Cl, Br, I, and Y = H, F, OH. $^{15}$N chemical shifts were used to quantify the cooperativity [36].

Although we have excluded cations from this review, we would like to report our studies involving the lithium cation. One characterizing the F–Li$^+$–F lithium bonds [38]; a number of homo-dimer and hetero-dimer complexes were studied (H$_3$C–F–Li$^+\cdots$F$_2$, H$_3$C–F–Li$^+\cdots$F–H, Cl–F$\cdots$Li$^+\cdots$F–Cl, F$_2\cdots$Li$^+\cdots$F$_2$, $\ldots$) and the spin-spin coupling constants (SSCC) calculated. A different approach was used to study the 1:1 and 2:1 complexes between hydrogen peroxide and its methyl derivatives with lithium cation in order to find if a huge static homogeneous electric field perpendicular to the magnetic field of the NMR spectrometer is able to differentiate enantiomers [39].

## 3. Alkaline Earth Bonds

Initially, this topic started with *Beryllium bonds* [40,41] and further extended to magnesium and calcium bonds along Group 2. Kollman, Liebman, and Allen suggested, in 1970, studying H$_2$Be$\cdots$OH$_2$, while they explained that HBeF is isoelectronic to HCN [34]. We contributed to this topic starting with a paper of 2009 entitled "Beryllium bonds, do they exist?" [42]. There, we noted that inorganic chemists have described BeX$_2$L$_2$ compounds in which X = F, Cl, Br, and L = NH$_3$ and other Lewis bases (for more recent papers concerning these complexes, see [43,44], and note that they do not call them beryllium bonds).

Beryllium bonds can modulate the strength of HBs (cooperativity) [45], transform azoles into gas-phase superacids [46], create σ-holes in molecules that are devoid of them (like CH$_3$OF) [47], spontaneous production of radicals [48], beryllium based anion sponges [49], etc.

Magnesium bonds were explored later on. Thus, Q. Li et al. studied the H$_2$NLi$\cdots$HMgX complexes where X = H, F. Cl, Br, CH$_3$, OH and NH$_2$ that are stabilized though a combination of magnesium and lithium bonds [50]. Scheiner et al. reported the effect of magnesium bonds on the competition between hydrogen and halogen bonds [51]. Montero-Campillo et al. discussed the synergy between tetrel bonds and alkaline earth bonds resulting in weak interactions getting strong [13]. Although NCI are generally studied in intermolecular complexes, there is a paper describing intramolecular magnesium bonds in malonaldehyde-like systems [52].

High-level calculations, using the complete basis set (CBS) extrapolation [CCSD(T)/CBS] of B$\cdots$BeR$_2$ and B$\cdots$MgR$_2$ complexes were carried out where B is a LB and R = F, H and CH$_3$ [53]. The Mg series show smaller electrophilicities than the Be series.

Finally, calcium bonds were studied in comparison with beryllium and magnesium bonds at producing huge acidity enhancements [54].

Although some authors have started calling them alkaline earth bonds [13,54], its use has still not become the norm.

## 4. Regium Bonds

This name (they are also called *Metal Coinage Bonds*) [55–57] is usually given to Group 11; we propose to include also group 10 (Ni, Pd, Pt). We cited Pt (group 10), Co, Rh, and Ir (group 9) in a paper on regium bonds [55], but nobody reports these systems as NCIs.

It is necessary to clearly differentiate clusters (e.g., Au$_2$ or Ag$_{11}$) (Figure 3) [58] from molecules (e.g., AuX) [59,60]. Brinck and Stenlid, based on their study of nanoclusters of Cu, Au, Pd, Pt, Rh, $\ldots$ ),

proposed a division of σ-holes, depending on the molecular electrostatic potential, into $\sigma_s$, $\sigma_p$, and $\sigma_d$-holes [61,62].

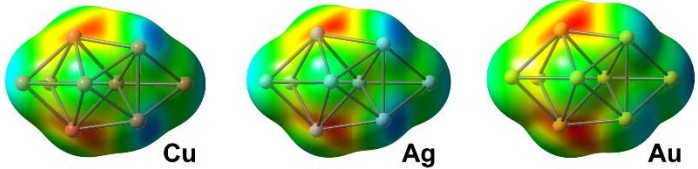

**Figure 3.** Coinage metal clusters [55].

The higher the oxidation degree (for instance, Au(III) vs. Au(I)) the more acidic the Lewis acid; see, for instance, the complex $(CF_3)_3Au\cdots$ pyridine [63]. We cited Legon in a 2014 paper [64] but did not define the Cl–Ag$\cdots C_2H_2$ complex as a regium bond (Figure 4):

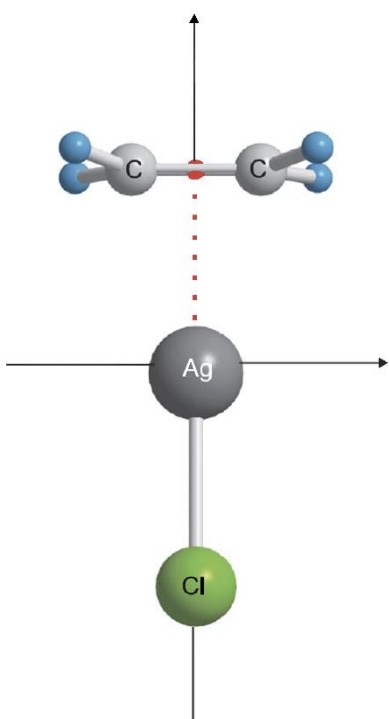

**Figure 4.** Experimental microwave (MW) structure of complex $C_2H_4\cdots$ Ag–Cl.

In 2019, several papers were published on regium bonds, from which we have selected the following four Reference works [65–68].

A comparative study of the regium and hydrogen bonds in $Au_2$:HX complexes was carried out at CCSD(T) level. In all cases, the regium bond complexes are more stable than HB ones. The binding energies for regium bonds complexes range between –24 and –180 kJ·mol$^{-1}$, whereas those of the HB complexes are between –6 and –19 kJ·mol$^{-1}$ [65]. Similarly, triel and regium bonds were compared, in particular they augmentative and diminutive interactions; the calculations were carried out at second order Møller-Plesset (MP2) perturbation theory [66]. For Cu, Ag, and Au atoms, the aug-cc-pVDZ-PP pseudopotential was used to account for relativistic effects.

A recent investigation described in detail the synthesis, X-ray characterization, and regium bonding interactions in a trichlorido-(1-hexylcytosine)gold(III) complex [67]. Moreover, this study also included an interesting search in the CSD, revealing that this type of noncovalent interaction is recurrent in X-ray structures and has remained essentially unobserved because of the underestimated van der Waals radius value tabulated for gold. Figure 5 shows the self-assembled dimer that is formed

in the solid state of trichlorido-(1-hexylcytosine)gold(III) where two symmetrically equivalent Au···Cl regium bonds are established.

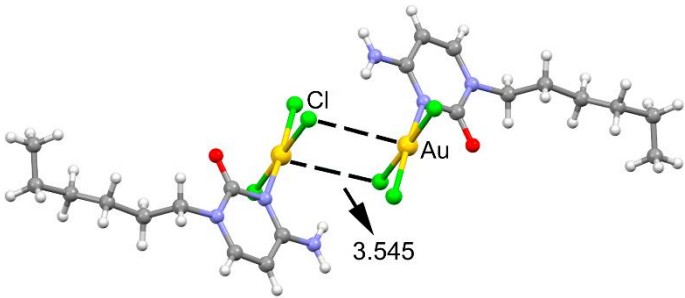

**Figure 5.** Self-assembled dimer of trichlorido-(1-hexylcytosine)gold(III) complex. Distance in Å.

Finally, regium bonds formed by MX (M = Cu, Ag, Au; X = F, Cl, Br) with phosphine-oxide and phosphinous acid were studied comparing oxygen-shared and phosphine-shared complexes. These complexes were investigated by means of ab initio MP2/aug-cc-pVTZ method [68].

A comparative study of the Lewis acidities of gold(I) and gold (III), specifically ClAu and $Cl_3Au$, towards different ligands (H, C, N, O, P, S) was carried out at the CCSD(T)/CBS level (an example of N base is given in Figure 6) [69]. The dissociation energies of the complexes are consistent with Yamamoto model. This author, in three fundamental papers [70–72], signaled that $AuCl_3$ behaves preferably as a σ-electrophilic Lewis acid with a $\eta^1$ hapticity typically towards heteroatom lone pairs, while AuCl behaves a π-electrophilic Lewis acid with a $\eta^2$ hapticity typically towards CC double and triple bonds. Amongst the unexpected findings is that both chlorides open the cyclopropane ring to afford a four-membered metallacycle and that the benzene complexes can show metallotropic shifts. Theoretical [73] and experimental [74] papers related to gold-arene structures have been published. Clearly, this field is one of higher growth in recent times.

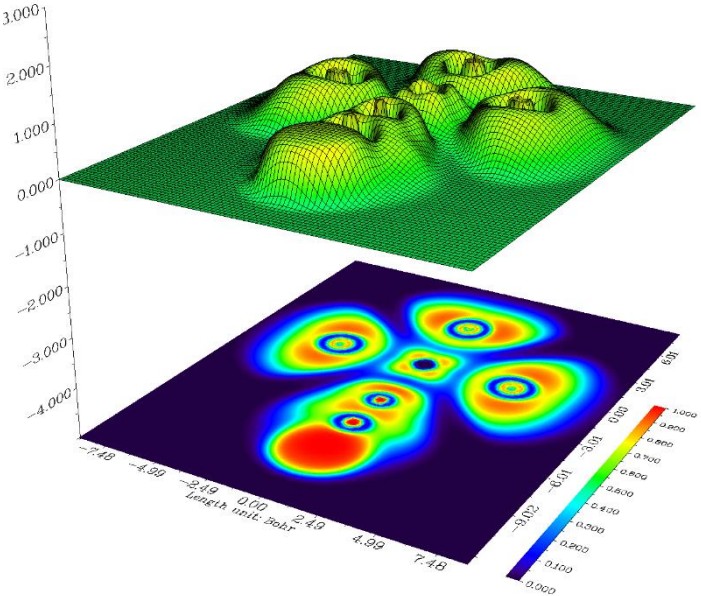

**Figure 6.** Electron localization function (ELF) analysis of the $Cl_3Au$···NCH complex.

The nature of the Au–N bond in Au(III) complexes with aromatic heterocycles led Radenkovic et al. to the conclusion that they have higher electrostatic than covalent character [75]. AIM analysis shows that the charge density of the Au–N bond is depleted along the bond path.

### 5. Spodium Bonds

As aforementioned, for elements of group 11 acting as electron acceptors, the name of regium bonds was proposed to define their interaction with Lewis bases. However, for the adjacent Group 12, the trivial name has not been yet defined. We propose herein to name these bonds "spodium bonds" because a derivative of the first element of the group (ZnO) is called spodium in Latin. It is important to emphasize that the interesting and remarkable work of Joy and Jemmis [76] anticipated that metals of the twelfth group might also participate in noncovalent interactions as Lewis acids. Moreover, these authors also showed that for groups 3–10, this type of interaction (denoted generically as metal bonding) is very scarce. In fact, they searched the Cambridge Structural Database (CSD) [77] and could not find any standard 18-electron transition-metal complexes where the metal participates in a weak interaction of type X−M⋯:A (A = Lewis Base).

The lack of σ-hole bonding (or metal bonding) in groups 3–10 is due to the fact that the possible σ-hole on the metal center is screened by the core electrons and diminished charge polarization. This is explained by the minimal orbital coefficient on the LUMO in the R–M bond (M belonging to groups 3–10). However, for metal complexes of elements of groups 11 and 12 (fully filled d orbitals), highly diffused valence s and p orbitals can sustain the σ-hole and they are capable to form M–bonds just like the main-group compounds. One of the first manuscripts describing spodium bonds was published by Chieh in 1977 [78]. It corresponds to a dichloro-bis(thiosemicarbazide)-mercury(II) complex that establishes highly directional spodium bonds. It can be clearly observed in Figure 7 that this compound forms in the solid state infinite 1D supramolecular chains where the electron donor (chlorido ligand) is located opposite to the polarized Hg–Cl bond at a distance of 3.25 Å that is slightly shorter than the sum of van der Waals radii (3.30 Å) and significantly longer than the sum of covalent radii (2.39 Å), thus evidencing the noncovalent nature of the interaction.

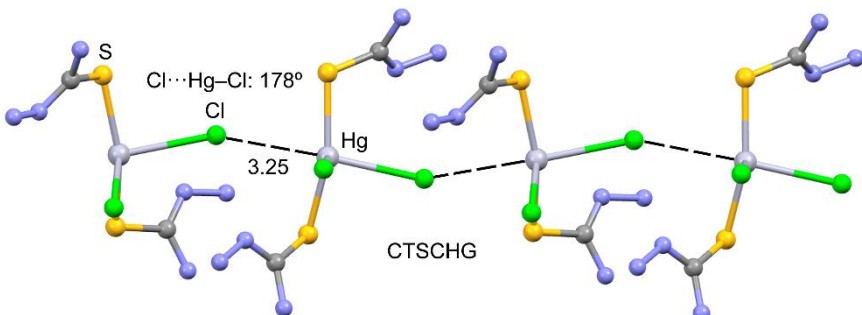

**Figure 7.** Spodium complexes of ZnCl$_2$. Distances in Å. The CSD reference code is indicated.

The nature of the metal⋯CO bonds in Group 12 metal carbonyl cations was analyzed by Frenking et al. [79] by studying the geometric and energetic features of their carbonyl complexes, which were also characterized using several computational tools like NBO and distribution of electron density. They showed that in Group 12 the M–CO bond strength in [M(CO)$_n$]$^{2+}$ complexes exhibits the trend Zn$^{2+}$ > Hg$^{2+}$ > Cd$^{2+}$ and, interestingly, the bond energies are strong for n = 1, 2, moderate for n = 3, 4, and weak for n = 5, 6. Moreover, they showed that Group 12 carbonyls [M(CO)$_n$]$^{2+}$ exhibit mainly coulombic attraction with quite small covalent contributions apart from [Hg(CO)]$^{2+}$ and [Hg(CO)$_2$]$^{2+}$ complexes. In contrast, covalent contributions were shown to be significant in the metal carbonyls of Group 11.

It is worthy to highlight the investigation by Vargas et al. where the synthesis and X-ray characterization of unprecedented monomeric 16-electron π-diborene complexes of Zn(II) and Cd(II) are reported, which are good examples of noncovalent spodium bonds [80]. As a matter of fact, stable π-complexes of d$^{10}$ transition metals like copper(I) and nickel(0) with olefins are known. However, such complexes involving d$^{10}$ Zn(II) are not known because the bond is too weak to generate isolable

compounds. This fact was explained taking into consideration the limited capacity of elements of Group-12 for π-back-donation. Vargas et al. overcame this drawback by using neutral diborenes because this type of compounds exhibits a high-lying π(B=B) HOMO orbital. In fact, they were able to synthesize in good yields M(II)-diborene (M = Zn, Cd) π-complexes. In addition to their X-ray characterization in the solid state, they were also detected in solution by NMR and UV-visible absorption spectroscopy. The M(II) centers are located over the center of the B=B bond and adopt a trigonal planar geometry almost equidistant to both boron atoms.

## 6. Triel Bonds

The name of triel bonding was proposed by Grabowski [81] in 2014 to describe the noncovalent interactions between elements of group 13 and electron rich atoms. However, the LA ability of triel atoms has been known for a long time [82–87]. In fact, trivalent triel compounds, such as trihydrides and trihalides, present a strong π-hole due to the empty p orbital, which is perpendicular to the plane of the molecule. This empty p-orbital determines the high directionality of the triel bonding. Since 2014, a number of experimental and theoretical studies have been published devoted to the study of the triel bond and its relation to reactivity [88–93]. As an example, in Figure 8, we show the X-ray structure of the hydrochloride of 4-pyridinylboronic acid, where the anion is located precisely over the B-atom in line with the location of the π-hole, as shown in the MEP surface (see Figure 8).

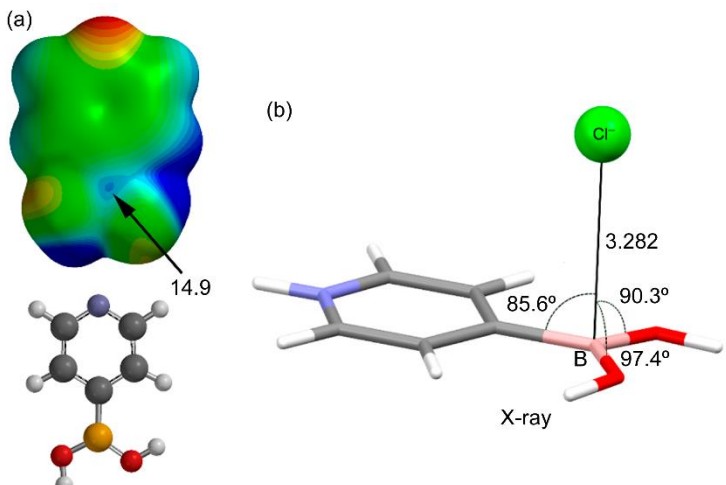

**Figure 8.** (**a**) Molecular electrostatic (MEP) surfaces of 4-pyridinylboronic acid with indication of the MEP value at the π-hole in kcal·mol$^{-1}$. (**b**) X-ray structure of the hydrochloride of 4-pyridinylboronic acid. The anion is located over the π hole at the boron atom. Distances in Å.

Energetically, the triel bond is very strong and presents highly covalent character. Actually, Leopold et al. [94] have named these type of complexes as "partially bonded complexes" after performing a systematic investigation on the geometric features of triel bonding complexes. The equilibrium distances are intermediate between van der Waals contacts and covalent bonds. It is interesting to highlight the behavior of triel bonds depending on the state. For instance, the triel bonding complex between F$_3$B and acetonitrile exhibits a B···N distances that is 2.01 Å in the gas phase and 1.63 A in the solid state due to cooperativity effects [95].

As a matter of fact, a significant attention has been paid to synergetic effects between triel bonds and a great deal of interactions, including hydrogen bonding [96], and other σ-hole based interactions in elements of group 17 [97], group 16 [98], group 15 [99], group 14 [100], and even regium bonding [66]. In these type of complexes, where two or more interactions coexist, the triel bond is usually the most favored one. Upon formation of the complex, the trivalent triel atom usually suffers a large deformation, changing its planar structure to a pseudo-tetrahedral one thus changing to an

sp$^3$-hybridization. Recently, 'like-like' In(III)···In(III) interactions was studied by Echeverría [101,102] in the crystal of trimethyltriphenyl-phosphine-indium. These unprecedented metallophilic interactions have not been described for the lighter elements of group 13.

## 7. Tetrel Bonds

A tetrel bond [103] was defined as a noncovalent bond between any electron donating moiety and a LA atom belonging to Group 14 of elements. The initial investigations were basically theoretical; [104–110] however, experimental research on TrB has rapidly grown in the last decade. Actually, there are plenty of examples in the literature reporting experimental [111,112] investigations on tetrel bonding, which was named as such in 2013 [113–116]. A differential feature of tetrel bonding compared to halogen, chalcogen and pnictogen bonding interactions is that the charge density distribution on the tetrel atom is not anisotropic (absence of lone pairs). Moreover, it should be emphasized that the accessibility of the σ-holes is reduced in tetrels because they are located in the middle of three sp$^3$-hybridized bonds. The behavior of carbon (also named carbon bonding in some studies) [111] is usually different because the rest of tetrels has a strong tendency to expand their valence. Indeed, the heavier tetrels tin and lead, which are commonly seen as metals, have rich coordination chemistry [117–120]. Furthermore, hypervalent species of silicon and germanium are very common [121–131]. Nevertheless, the heavier tetrel atoms (Ge–Pb) participate in noncovalent tetrel bonding interactions when they are in a chemical context avoiding hypervalency, see for instance the SiO$_{12}$(OH)$_8$ cage in Figure 9 [132,133]. In fact, since the atomic polarizability increases in a given group of the periodic table on going from lighter to heavier elements, the stronger interactions in this group are expected for tin and lead [134–136].

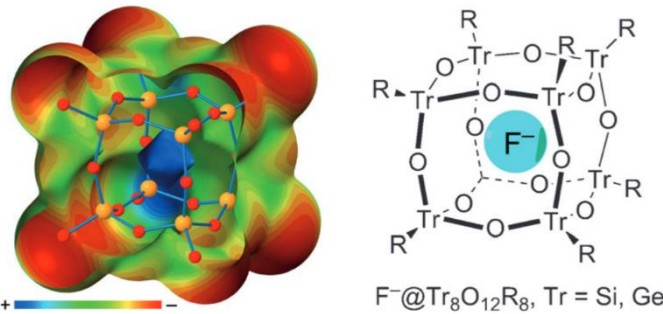

F$^-$@Tr$_8$O$_{12}$R$_8$, Tr = Si, Ge

**Figure 9.** Left: Molecular electrostatic potential open surface of SiO$_{12}$(OH)$_8$ cage. Right: A F$^-$ ion encapsulated inside a Tr$_8$O$_{12}$R$_8$ cage reported by Bauzá et al. [104].

For carbon, tetrel complex can also be understood as the start, [A:···CR$_3$A'] or outcome, [ATrR$_3$···:A'], of an S$_N$2 nucleophilic attack [105] being the transition state an hypervalent specie. Most of the works on tetrel bonding focus on the heavier atoms leaving "carbon bonding" mostly unstudied. In an sp$^3$ hybridized electron deficient C atom, such as CF$_4$, there is only a limited space available for the LB to interact with C due to its small size. In addition, LB gets very close to negative electrostatic potential of F in CF$_4$. Frontera et al. [107] showed both theoretically and experimentally searching the CSD [77] that a convenient way to expose the σ-hole is to use cyclic X$_2$C–CX$_2$ structures (X = F, CN) where the accessibility of the σ-hole increases as the size of the cycle decreases. In fact, the (CN)$_2$C–C(CN)$_2$ motif was found to be highly directional in 1,1',2,2'-tetracyanocyclopropane/cyclobutane structures.

When sp$^2$-hybridized electron deficient C-atoms are considered (π-hole instead of σ-hole), the accessibility is not a problem. In this sense, pioneering π-hole interactions were described in 1973 by Bürgi and Dunitz [137,138] in a series of X-ray structural analyses disclosing the trajectory along a LB or nucleophile predominantly attacks the π-hole of a C=O. More than 20 years later, Egli and co-workers described the ability of guanosine to interact with the LBs (O-atom of de-oxiribose) and its importance in the stabilization of Z-DNA form [139].

## 8. Pnictogen Bonds

These bonds were first described in 2011 in three papers, one experimental [140] and two theoretical [141,142]. An authoritative review was published in a book by some of us (Chapter 8: J. E. Del Bene, I. Alkorta, J. Elguero, The Pnicogen Bond in Review: Structures, Binding Energies, Bonding Properties, and Spin-Spin Coupling Constants of Complexes Stabilized by Pnicogen Bonds, 191-264) [3,143], and another by Scheiner [144]. Although pnictogen bonds are, after halogen bonds, the most studied weak interaction, these bonds have been treated in a reduced number of books and reviews [103,143]. Grabowski classified them as tetrahedral Lewis acid centers [103]. Legon discussed these bonds in an article called " Tetrel, pnictogen and chalcogen bonds identified in the gas phase before they had names: a systematic look at noncovalent interactions" [57].

They are also called "pnicogen bonds" but the pnictogen name should prevail. Similar to halogen bond, pnictogen bond is also a noncovalent interaction. In pnictogen-bond complex, pnictogen atoms (Group VA elements) act as Lewis acid, which can accept electrons from electron donor groups.

Legon pointed out that tetrel, pnictogen, and chalcogen bonds were known in the gas phase (mainly by this author, using rotational spectroscopy) before they had names [56]. Recently, the gas-phase structure of a pnictogen-bonded compound was determined (Figure 10) [145].

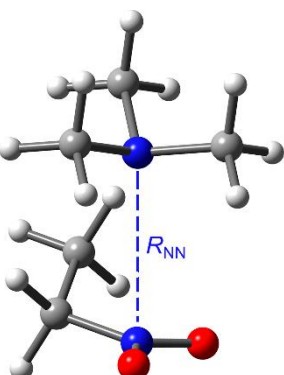

**Figure 10.** The nitromethane···trimethylamine pnictogen complex [145].

One of our main contributions to pnictogen bonds are the EOM/CCSD calculations, made by J. E. Del Bene, of $^{31}P$ coupling constants through the pnictogen bond, we called $^{np}J(X-^{31}P)$ [142]. Of our papers concerning pnictogen bonds, we have selected the following eight ones [146–153]. Most of these papers were calculated at the MP2/aug'-cc-pVTZ basis set. We and others have found that $FPH_2$ and related $YPH_2$ (Y = H, OH, $OCH_3$, $CH_3$, $NH_2$) and $FH_2X$ (X = P, As) are strong and directional Lewis acid especially suited for theoretical studies [154–156]. Highly acidic heteroboranes yield strong pnictogen bonds [157].

Li, McDowell et al. have shown that upon protonation, the binding distance of the pyridine-(4)-$PH_2$···$NH_3$ & $PH_3$ complexes becomes shorter and the interaction energy is more negative. This shows that the pnictogen bond is strengthened by the protonation of the N atom of pyridine [158]. P···$\pi$ and $\pi$-hole pnictogen bonds have been studied [159,160] and the $Cl_3P$···$C_6H_6$ complex studied experimentally by FTIR spectroscopy (Figure 11) [159]. Two important papers have been published, one on the catalysis by pnictogen bonds where there is a distinction between $PH_2F$ $\sigma$-hole vs. $PO_2F$ $\pi$-hole [161], and the other of supramolecular structures using triple pnictogen bonds [162].

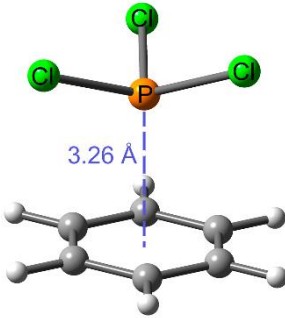

**Figure 11.** The $Cl_3P\cdots$benzene complex [159].

Complexes $H_2XP\cdots NXH_2$ (X = H, $CH_3$, $NH_2$, OH, F, Cl) presenting $P\cdots N$ pnictogen bonds show stabilization energies between 8 and 39 kJ·mol$^{-1}$ [146]. $^{31}P$ chemical shieldings and $^{1p}J$(N-P) SSCC across the pnictogen interaction were calculated. The last ones exhibit a quadratic dependence on the N–P distance for complexes $H_2XP\cdots NXH_2$, similar to the dependence of $^{2h}J$(X–Y) on the X–Y distance for complexes with X–H$\cdots$Y hydrogen bonds.

The study the influence of F–H$\cdots$F hydrogen bonds on the $P\cdots P$ pnictogen bond in complexes $n$FH$\cdots$(PH$_2$F)$_2$ for $n = 1-3$ shows that the formation of F–H$\cdots$F hydrogen bonds leads to a shortening of the P–P distance, a lengthening of the P–F distance involved in the hydrogen bond, a strengthening of the $P\cdots P$ interaction, and changes in atomic populations [147]. $^{31}P$ chemical shieldings, and $^{1p}J$ (P–P) coupling constants were calculated.

Pnictogen-bonded cyclic trimers (PH$_2$X)$_3$ with X = F, Cl, OH, NC, CN, CH$_3$, H, and BH$_2$ have been computed (Figure 12) [148]. Most of these complexes have $C_{3h}$ symmetry and binding energies between −17 and −63 kJ·mol$^{-1}$. The NMR properties of chemical shielding and $^{31}P$–$^{31}P$ coupling constants have also been evaluated.

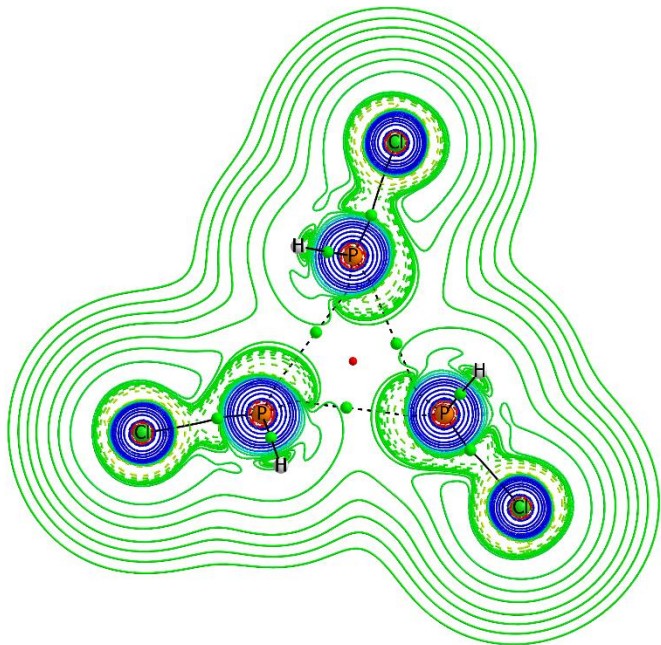

**Figure 12.** Laplacian of the electron density on the molecular plane of the trimer of (PH$_2$Cl) [148].

Three papers have been reported comparative studies of different NCIs. In the first one [149]. the influence of substituent effects on the formation of $P\cdots Cl$ pnictogen bonds or halogen bonds was assessed. There, the potential energy surfaces $H_2FP\cdots ClY$ for Y = F, NC, Cl, CN, CCH, CH$_3$, and H were explored finding three different types of halogen-bonded complexes with traditional,

chlorine-shared, and ion-pair bonds. Two different pnictogen-bonded complexes have also been found on these surfaces. In the second one [153], ab initio calculations were carried out in search of equilibrium dimers on $(XCP)_2$ potential energy surfaces, for X = CN, Cl, F, and H. Five equilibrium dimers with $D_{\infty h}$, $C_{\infty v}$, $C_s$, $C_{2h}$, and $C_2$ symmetries exist on the $(ClCP)_2$ potential energy surface, four on the $(FCP)_2$ and $(HCP)_2$ surfaces, and three on the $(NCCP)_2$ surface. These dimers are stabilized by traditional halogen, pnictogen, and tetrel bonds, and one of them by a hydrogen bond. Finally, Resnati et al. reported an example of a cocrystal where a pnictogen bond prevails over halogen and hydrogen bonds [163].

Another paper reported studies on P(V) complexes [150]. Pnictogen-bonded complexes $H_nF_{5-n}P\cdots N$-Base, for $n$ = 0–5 were studied (two illustrative examples are given in Figure 13). The computed distances and $F_{ax}-P-F_{eq}$ angles in complexes $F_5P$:N-base are consistent with experimental CSD data [77]. All of the complexes with $PF_5$, $PHF_4$, $PH_4F$, and $PH_5$ have $C_{4v}$ symmetry, which is the same symmetry as that of the Berry transition structures of the monomers which lead to the exchange of axial and equatorial atoms.

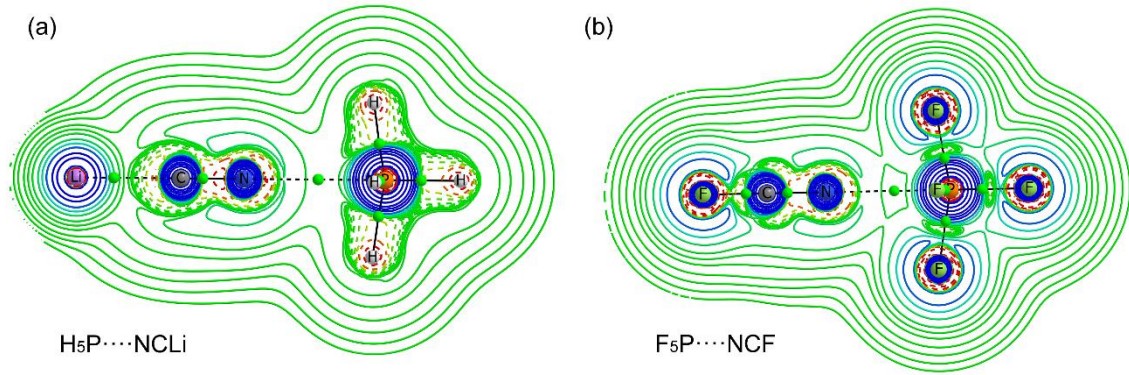

**Figure 13.** (**a**) Laplacian of the electron density on the molecular plane of $H_5P\cdots NCLi$ complex; (**b**) $F_5P\cdots NCF$ complex [150].

An ab initio study of the hydration process of metaphosphoric acid shows the importance of the pnictogen interactions [151]. This work was carried out at the MP2/6-31+G(*d,p*) and MP2/aug-cc-pVTZ computational levels. Up to three explicit water molecules have been considered. The inclusion of more than one water molecule produces important cooperative effects and a shortening of the $O\cdots P$ pnictogen interaction simultaneously the reaction barrier drops about 50 kJ mol$^{-1}$.

A general study of several kinds of NCIs was carried at the MP2/aug-cc-pVTZ computational level. In this paper [152], the dissociation energies $D_e$ of 250 complexes $B\cdots A$ composed of 11 Lewis bases B ($N_2$, CO, HC≡CH, $CH_2$=$CH_2$, $C_3H_6$, $PH_3$, $H_2S$, HCN, $H_2O$, $H_2CO$, and $NH_3$) and 23 Lewis acids (HF, HCl, HBr, HC≡CH, HCN, $H_2O$, $F_2$, $Cl_2$, $Br_2$, ClF, BrCl, $H_3SiF$, $H_3GeF$, $F_2CO$, $CO_2$, $N_2O$, $NO_2F$, $PH_2F$, $AsH_2F$, $SO_2$, $SeO_2$, $SF_2$, and $SeF_2$) can be represented to good approximation by means of the equation $D_e = c'N_BE_A$, in which $N_B$ is a numerical nucleophilicity assigned to B, $E_A$ is a numerical electrophilicity assigned to A, and $c'$ is a constant, conveniently chosen to have the value 1.00 kJ mol$^{-1}$. The 250 complexes were chosen to cover a wide range of noncovalent interaction types, namely: (1) the hydrogen bond; (2) the halogen bond; (3) the tetrel bond; (4) the pnictogen bond; and (5) the chalcogen bond.

Diederich orthogonal interactions ($N:\cdots O_2N$) are pnictogen bonds when there is a nitrogen lone pair acting as the Lewis base and a nitrogen atom of the nitro group acting as the Lewis acid [164–166]. These interactions have been used by us [167–170] and by others to explain some experimental observations [171]. A theoretical paper entitled "Orthogonal interactions between nitryl derivatives and electron donors: pnictogen bonds"; in this paper complexes from nitryl derivatives ($NO_2X$, X = CN, F, Cl, Br, $NO_2$, OH, CCH, and $C_2H_3$) and molecules acting as Lewis bases ($H_2O$, $H_3N$,

CO, HCN, HNC and HCCH) have been obtained at the MP2/aug-cc-pVTZ computational level; a search in the CSD database [77], was carried out, showing a large number of similar interactions in crystallographic structures.

## 9. Chalcogen Bonds

These bonds have received less attention than the pnictogen bonds, probably due to the fact that P is in chemistry and in biochemistry more important than S. In addition, note that $^{31}$P is a very good nucleus for NMR (spin 1/2, natural abundance 100%) and $^{33}$S a "bad" one (spin 3/2, natural abundance 0.76%). For books and reviews on chalcogen bonds, see [172–175].

The name "chalcogen bond" was introduced in 2009 by Wang, Ji and Zhang [176]. But papers discussing these NCIs were long known [177–181]. In particular, Gleiter et al. [181] investigated the intermolecular interactions between two molecules containing group 16 elements. The strength of this interaction increases steadily when going from O via S to Se and reaches its maximum for Te. Addition of electron-withdrawing substituents increases the strength of the bond. S···S contacts in thioamides have been studied both experimentally (charge densities) and theoretically [182].

Since most molecules have several kinds of atoms, and since all atoms can be Lewis acids, then, confronted with a Lewis base, several types of NCIs can be formed. For this reason, many papers have been devoted to the competition between some combination of hydrogen, alkaline-earth, tetrel, pnictogen, chalcogen, and halogen bonds [157,183–190]. Curiously, although the nature of the base can change the nature of the most stable acid, none of these publications reported an inversion of acidity. Huynh electronic parameter and its correlation with Hammett σ constants were determined for neutral chalcogen donors [187].

More interesting are the papers reporting cooperative (augmentative) effects where a NCI is reinforced by another NCI, to the point to reach extraordinary values of gas-phase acidity or basicity [191,192].

Although most chalcogen bonds are related to intermolecular situations, a few correspond to intramolecular situations, e.g., to 1,8-disubstituted naphthalenes [193,194]. Other interesting topics related to chalcogen bonds are their use in chiral recognition [195], chalcogen-bonding catalysis [196], and the use by Diederich of benzo[*c*][1,2,5]thiadiazoles and benzo[*c*][1,2,5]telluradiazoles to build up capsule dimers [197], followed by a study of "2S-2N" squares formed by benzo[*c*][1,2,5]thiadiazoles [198].

## 10. Halogen Bonds

Halogen bonding is a σ-hole interaction of type R–X···:A (X = any element of group 17 including astatine [199]); that is currently experiencing a significant interest in the field of supramolecular chemistry [200–204]. It is the most directional interaction [205] of the σ-hole family, and it can be easily tuned by selecting the type of halogen atom involved (X = I > Br > Cl >> F) [206,207] and nature of the substituent R. This tunability facilitates the rational design of X-bonded catalysts [208,209] and supramolecular synthons to be utilized in crystal engineering [210–212]. The distribution of the electron density in a covalently bonded halogen atom is anisotropic. That is, it shows a region of positive electrostatic potential [213] along the extension of the covalent bond that confers it the ability to act as Lewis acid (i.e., halogen bond donor) [214]. Moreover, it also has a region of negative potential (negative belt) associated to the electron lone pairs conferring it the possibility to act as an electron-rich halogen bond acceptor (Lewis base) [215]. Recently, the X–Bond interaction was used in the field of molecular machines [216–218] providing a new dimension to this interaction. In addition, regarding its counterpart (Lewis base), it was recently demonstrated that transition metal complexes can act as halogen bond acceptors [219–221]. Clark [222,223] and Hobza [224,225] related the strength of halogen bonding to the so-called "polar flattening".

Several excellent reviews [181,201,202,226] and books [33,227] are available in the literature describing most aspects of halogen bonding; therefore, only some general features are commented

herein briefly. Halogen bonding is comparable in strength [228] to the ubiquitous hydrogen bond, however, more sensitive to steric effects because the σ-hole is located in a small region of the van der Waals surface along the extension of the R–X bond. A differentiating feature is that the H-bond can be only tuned varying the nature of R and the halogen bond can be tuned varying both R and X. The nature of the X-bond is still under discussion in the literature [229,230]. Note that most theoretical studies propose that an important contribution comes from the stabilization due to donor-acceptor orbital interactions. That is, a filled π or n orbital from the Lewis basic site donates electron density to the antibonding R–X sigma bond orbital [231–233]. Other important contributions are electrostatic effects, polarization in heavy halogens, and dispersion forces that depend upon on the nature of both the Lewis acid and Lewis base [234]. Finally, Kozuch and Martin used halogen bonds as benchmarks for theoretical analyses of wave methods and DFT methods [235].

## 11. Aerogen Bonds

A noble gas (or aerogen) [236] bond (NgB) was recently defined as: *the interaction between an electron rich atom or group of atoms and any element of Group-18 acting as electron acceptor* [237]. While reports on π,σ-hole interactions involving atoms of groups 14 to 17 as LA have exponentially grown in recent years, investigations on experimental aerogen bonding are scarce. One of those was reported by Schrobilgen's group [238], where they synthesized and X-ray characterized several xenon salts [N(C$_2$H$_5$)$_4$]$_3$ [X$_3$(XeO$_3$)$_3$] X = Cl, Br. These salts form three aerogen bonding interactions with the halides by using the three σ-holes opposite to the O=Xe bonds. Similar behavior was observed by Goettel et al. [239] in their investigation of a series of XeO$_3$ adducts with nitriles since they also form three aerogen bonds in the solid state.

In Figure 14, two X-ray structures are represented where the XeO$_3$ establishes three concurrent aerogen bonds with pyridine N-atoms [232]. These aerogen bonds are shorter for the *p*-dimethylaminopyridine Lewis base due to its stronger basicity compared to pyridine.

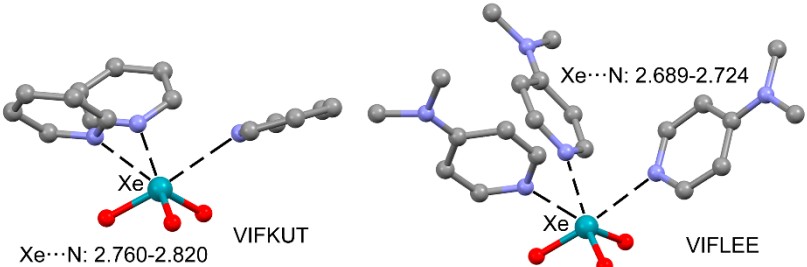

**Figure 14.** Aerogen bonging interactions in two XeO$_3$ adducts (CSD refcodes VIFKUT and VIFLEE [77]). Aerogen bonds are represented as black dashed lines.

Britvin et al. also demonstrated the tendency of xenon(VI) to form oxide structures synthesizing K$_4$Xe$_3$O$_{12}$, an unprecedented perovskite based on xenon. Its importance is due to the fact that xenon is the only p-block element that forms perovskite frameworks by using a single cation (K$^+$). Remarkably, the authors showed that aerogen bonds are the NCIs that preserve the structural integrity of the perovskite. It is interesting to highlight that these compounds are explosive and the aerogen bonds have been proposed to be the trigger bonds responsible for the detonation [240,241].

Several computational works studied this interaction energetically and geometrically, including its physical insights [242–253]. Interestingly, the effect of increasing the pressure (up to 50 GPa) on the aerogen interactions in XeO$_3$ was also analyzed, resulting in O-hopping along the noncovalent Xe–O···Xe aerogen bonds, resembling H-hopping commonly observed in hydrogen bonds [254]. Moreover, cooperativity effects in aerogen bonding clusters were studied [255] and the interplay with other interactions, as well [256–259].

## 12. Other Bonds

Cation-π and anion-π (or lone pair-π) [260,261] and even π-π stacking between a π-excessive and π-deficient aromatic rings (Figure 15) can be classified as LA/LB complexes (the Lewis acids being the cation and the hexafluorobenzene and the Lewis bases the anion, the lone pair and hexamethylbenzene) could be classified as tetrel bond since the carbon atoms act like LA (in the case of $C^+$ it depends on its nature, i.e., C = Na should be an alkaline bond). However, we have decided not to force our systematization running against practices shared by the scientific community.

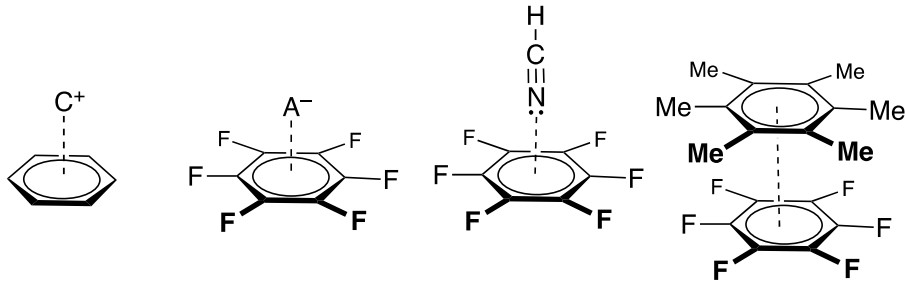

**Figure 15.** Cation-π, anion-π, lone pair–π and π-π stacking.

## 13. Modeling

The use of statistical methods to establish extra-thermodynamic relationships [262] for discussing values obtained by quantum methods presents the problem that they have no error, unlike experimental values; note that, without error, no statistical methods can be applied. In spite of this flaw, regression analysis is currently applied to values without error [263].

Two kinds of models are most commonly used: geometrical models like the Hammett, Taft, Grunwald-Winstein equations and the models subjacent to the HSAB (Hard Soft Acid Base)principle. Since we are dealing with Lewis acids and bases, it would be interesting to write a quantitative model that corresponds to hard-hard and soft-soft interactions being strong and hard-soft/soft-hard being weak. We are aware of Mayr et al. criticism of HSAB [264] but note a paper of 2002 by Chandrakumar and Pal entitled "A systematic study on the reactivity of Lewis acid-Base complexes through the local Hard-Soft Acid-Base principle" [265] where they succeed in calculating correctly the interaction energy of complexes using a HSAB model (not cited by Mayr in 2011). A quantitative version of the HSAB principle is Drago's ECW model [266,267].

Alkorta and Legon in two papers, which are (i) "Nucleophilicities of Lewis bases B and electrophilicities of Lewis acids A determined from the dissociation energies of complexes B···A involving hydrogen bonds, tetrel bonds, pnictogen bonds, chalcogen bonds and halogen bonds" and (ii) "Noncovalent interactions involving alkaline-earth atoms and Lewis bases B: An ab initio investigation of beryllium and magnesium bonds, B···$MR_2$ (M = Be or Mg, and R = H, F or $CH_3$") use geometrical models to analyze $D_e$ (equilibrium dissociation energies) in function of $k_\sigma$ (quadratic force constants) or NB (nucleophilicity of the Lewis base, B) plus $E_A$ (electrophilicity of the Lewis acid): $D_e = a_0 + a_{ij}N_B E_A$ [53,152].

Steric effects are inexistent for protonation in the gas-phase due to the small size of the proton and appear in solution due to solvation, for example, by water molecules [268,269]. For HBs, steric effects have been found, but they are weak or inexistent [270–273]; on the other hand, steric effects are important in NCIs giving yield to a new concept, that of "Frustrated Lewis Pairs" (FLP) [274–278].

## 14. Application Con Cahn-Ingold-Prelog Rules to Complexes Formed by Weak Interactions (Including Hydrogen Bonds)

For all the situations where the Cahn-Ingold-Prelog priority rules apply for covalent and coordinative structures (ligancy four, axial, planar, ... ) [279,280], the priority rules also apply for noncovalent complexes [281,282]. This is particularly useful for crystal structures.

## 15. A General Definition for Weak Interactions (Including HBs)

A weak interaction between a Lewis acid and a Lewis base is established if the stabilizing forces (electrostatic, dipole-dipole, covalent, ... ) overcome the repulsion forces (steric). It is not necessary that the complex should be the lowest minimum; it suffices that there is a barrier between the complex and other minima of lower energy.

## 16. Summary and Outlook

The number and quality of recent references prove that NCIs are a topic of great and increasing interest. However, as the analysis of the authors of these references show, they belong to a reduced number of groups proving that NCIs are still not part of the large community of chemists. We hope this review will contribute to their diffusion and general acceptance.

A systematic naming resulting from identifying the interaction referring to the Group of the periodic table is very convenient for the sake of unambiguousness. Basically, all donor-acceptor noncovalent interactions can be identified by the element acting as the electrophile. This criterion has been already adopted by the IUPAC for the definition of hydrogen, halogen, and chalcogen bonds. This can be systematically applied to attractive interactions formed by the elements of Groups 1, 2, 10–18 and also to transition metals in a near future. Other names used in the literature like lithium bond, bromine bond or carbon bond can be considered sub-classes of alkali metal bond, halogen bond, and tetrel bond, respectively. Other interactions, like π–π stacking, lp–π, or anion–π interactions involving heteroaromatics, cannot be included in this systematic nomenclature. In contrast, the cation–π interaction could be classified using this nomenclature by using the name of the group to which the cationic element belongs.

It can be predicted that more gas-phase MW structures will be determined in a not so-distant future. Organometallic chemists will report new structures of the regium and spodium classes. Other future developments will be attached to the biological importance of the NCIs.

**Author Contributions:** Conceptualization, I.A., A.F. and J.E.; methodology, I.A., A.F. and J.E.; validation, I.A., A.F. and J.E.; writing—original draft preparation, I.A., A.F. and J.E.; writing—review and editing, I.A., A.F. and J.E. All authors have read and agreed to the published version of the manuscript.

**Funding:** This work was carried out with financial support from the Ministerio de Ciencia, Innovación y Universidades of Spain (PGC2018-094644-B-C22) and Comunidad Autónoma de Madrid (P2018/EMT-4329 AIRTEC-CM). Thanks are also given to the CTI (CSIC) for their continued computational support. AF thanks the MICIU/AEI (project CTQ2017-85821-R FEDER funds) for financial support.

**Acknowledgments:** This is dedicated to Anthony C. Legon, pioneer of studies of noncovalent interactions.

**Conflicts of Interest:** The authors declare no conflict of interest.

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
