# Peer review of "Not Only Hydrogen Bonds: Other Noncovalent Interactions"

_crystals, doi:10.3390/cryst10030180_

Round 1
Reviewer 1 Report
If the objective of the article is to be a review, this is a good one.
However, I find it lacking explanations of the characteristics of the bonds itself, which, to my viewpoint, bring many errors and exaggerations on the distinction of the bonds.
First of all, while the idea of “halogen bond” made sense, since this was the first type of bonds of this style, the extreme multiplication of names horrifies me. Chalcogen, aerogen, pnictogen, etc., are all the same type of bonds, and therefore they require a single name that will encompass them all. I have heard of “hole interactions”, which seems to be adequate, as all of these bonds are based on sigma or pi holes (even the H-bond). What they cannot be called is “non-covalent interactions”, since NCIs involve many more cases than the ones described here. It is an error to define NCI as the LA/LB interactions, when London forces are prototypical NCIs.
But more important, many of the bonds described here are really different. For instance, what is called here “triel bond” is nothing more than the boron trying to fill its octet. The fact that it is at 90 degrees does not make it a pi-hole interaction (as will appear in aldehydes, for instance). It is a big mistake to consider it as the same style of bond as hole interactions, where the octet is already full, but the LA comes from the opposed covalent bond. Similarly, alkali and alkaline-earth bonds are not, in any way, hole interactions. In the case of the Be bond, it resembles the boron bonds (unfilled octet). In the case of Li+ and other alkali, they are cations! In no way they can be considered in the same family as halogen bonds. Or, in the discussed case of carbones, they are not tetrel bonds, as they bare no similarities with them.
Furthermore, I do not like the name distinction of the bonds to transition metals. “Regium” and “Spodium”, as well as all the other elements of the middle of the periodic table, are transition metals. Why do we need to distinguish them according to the column if they all share the same characteristics (depending on the number of electrons)? But more important, systems as Cl3Au-NCH are not NCIs. Not at all. The Au-N bond distance is shorter than the sum of their covalent radii. These are the classical coordinative bonds of the Dewar–Chatt–Duncanson model. There is no need to change the nomenclature of such bonds when the change is erroneous. Same goes for the spodium bonds, when figure 7 clearly show that they have short bond lengths and they are not even formed at 180 or 90 degrees as sigma or pi holes require.
In conclusion, the article as a review is effective, covering a large amount of literature (beyond some papers that the authors might consider as well, see below). But as a critical analysis of the discussed bonds, it not only lacks information, it actually misrepresents the bonds, and it should be thoroughly fixed.
Extra articles to consider:
Chemical bonds:
Wang, C.; Danovich, D.; Mo, Y.; Shaik, S. On The Nature of the Halogen Bond. J. Chem. Theory Comput. 2014, 10 (9), 3726–3737. https://doi.org/10.1021/ct500422t.
Wang, C.; Guan, L.; Danovich, D.; Shaik, S.; Mo, Y. The Origins of the Directionality of Noncovalent Intermolecular Interactions. J. Comput. Chem. 2016, 37 , 34–45. https://doi.org/10.1002/jcc.23946.
Grabowski, S. J.; Sokalski, W. A. Are Various Sigma-Hole Bonds Steered by the Same Mechanisms? ChemPhysChem 2017. https://doi.org/10.1002/cphc.201700224.
Wang, H.; Wang, W.; Jin, W. J. σ-Hole Bond vs π-Hole Bond: A Comparison Based on Halogen Bond. Chem. Rev. 2016, 116 (9), 5072–5104. https://doi.org/10.1021/acs.chemrev.5b00527.
Reviews:
Edwards, A. J.; Mackenzie, C.; Spackman, P.; Jayatilaka, D.; Spackman, M. A. FDHALO17: Intermolecular Interactions in Molecular Crystals: What’s in a Name? Faraday Discuss. 2017. https://doi.org/10.1039/C7FD00072C.
Costa, P. J. The Halogen Bond: Nature and Applications. Physical Sciences Reviews 2017, 2 (11). https://doi.org/10.1515/psr-2017-0136.
Kozuch, S.; Martin, J. M. L. Halogen Bonds: Benchmarks and Theoretical Analysis. J. Chem. Theory Comput. 2013, 9 (4), 1918–1931. https://doi.org/10.1021/ct301064t.
Astatine:
Guo, N.; Maurice, R.; Teze, D.; Graton, J.; Champion, J.; Montavon, G.; Galland, N. Experimental and Computational Evidence of Halogen Bonds Involving Astatine. Nature Chemistry 2018, 10 (4), 428–434. https://doi.org/10.1038/s41557-018-0011-1.
Receptors:
Molina, P.; Zapata, F.; Caballero, A. Anion Recognition Strategies Based on Combined Noncovalent Interactions. Chem. Rev. 2017. https://doi.org/10.1021/acs.chemrev.6b00814.
Scheiner, S. FDHALO17: Comparison of Halide Receptors Based on H, Halogen, Chalcogen, Pnicogen, and Tetrel Bonds. Faraday Discuss. 2017. https://doi.org/10.1039/C7FD00043J.
Author Response
First of all we would like to thank this reviewer for his/her careful reading of the manuscript, suggestions, corrections. Our answers and corrections follow:
Comment: If the objective of the article is to be a review, this is a good one.
However, I find it lacking explanations of the characteristics of the bonds itself, which, to my viewpoint, bring many errors and exaggerations on the distinction of the bonds.
First of all, while the idea of “halogen bond” made sense, since this was the first type of bonds of this style, the extreme multiplication of names horrifies me. Chalcogen, aerogen, pnictogen, etc., are all the same type of bonds, and therefore they require a single name that will encompass them all. I have heard of “hole interactions”, which seems to be adequate, as all of these bonds are based on sigma or pi holes (even the H-bond). What they cannot be called is “non-covalent interactions”, since NCIs involve many more cases than the ones described here. It is an error to define NCI as the LA/LB interactions, when London forces are prototypical NCIs.
But more important, many of the bonds described here are really different. For instance, what is called here “triel bond” is nothing more than the boron trying to fill its octet. The fact that it is at 90 degrees does not make it a pi-hole interaction (as will appear in aldehydes, for instance). It is a big mistake to consider it as the same style of bond as hole interactions, where the octet is already full, but the LA comes from the opposed covalent bond. Similarly, alkali and alkaline-earth bonds are not, in any way, hole interactions. In the case of the Be bond, it resembles the boron bonds (unfilled octet). In the case of Li+ and other alkali, they are cations! In no way they can be considered in the same family as halogen bonds. Or, in the discussed case of carbones, they are not tetrel bonds, as they bare no similarities with them.
Answer:
Regarding the comment on triel bonds, please read the general answer below.
Alkali bonds and lithium bonds: we have modified the sentence to show that most of the reported examples concern organometallic compounds that in the gas phase correspond to neutral Li and Na atoms.
Concerning the comment that carbones are not tetrel bonds the reviewer is absolutely right; the sentence has been removed. We thank the reviewer to help us to remove this error.
Comment:
Furthermore, I do not like the name distinction of the bonds to transition metals. “Regium” and “Spodium”, as well as all the other elements of the middle of the periodic table, are transition metals. Why do we need to distinguish them according to the column if they all share the same characteristics (depending on the number of electrons)? But more important, systems as Cl3Au-NCH are not NCIs. Not at all. The Au-N bond distance is shorter than the sum of their covalent radii. These are the classical coordinative bonds of the Dewar–Chatt–Duncanson model. There is no need to change the nomenclature of such bonds when the change is erroneous. Same goes for the spodium bonds, when figure 7 clearly show that they have short bond lengths and they are not even formed at 180 or 90 degrees as sigma or pi holes require.
Answer:
We have used different examples to better illustrate the noncovalent nature of the interaction for spodium and regium bonds (see Figures 5 and 7).
According a paper (now included in the main text) The Au(III)···N bond is higher electrostatic than covalent (New J. Chem. 2017, 41, 12407). We have maintained the example adding this comment. It can be removed in a second revision round if the referee considers it mandatory.
Comment:
In conclusion, the article as a review is effective, covering a large amount of literature (beyond some papers that the authors might consider as well, see below). But as a critical analysis of the discussed bonds, it not only lacks information, it actually misrepresents the bonds, and it should be thoroughly fixed.
General answer. This is a review that reports literature results (including ours) with fidelity; if the authors describe a bond as a NCI, we reported it unchanged. The reviewer is horrified by the extreme multiplication of names, but all classification systems results in an increasing number of names. Using "hole interactions" for all of them is going against the general tendency. Note that all the extra articles to consider use most of the names: halogen, pnicogen, chalcogen, tetrel, ... bonds. Also, the fact that all these ten papers cite our publications indicates that our viewpoint is not strongly opposed by other authors.
We have added a general comment to explain that other views are possible.
Comment:
Extra articles to consider:
Answer:
All the ten articles have been included in the revised version.
Chemical bonds:
Wang, C.; Danovich, D.; Mo, Y.; Shaik, S. On The Nature of the Halogen Bond. J. Chem. Theory Comput. 2014, 10 (9), 3726–3737. https://doi.org/10.1021/ct500422t.
Wang, C.; Guan, L.; Danovich, D.; Shaik, S.; Mo, Y. The Origins of the Directionality of Noncovalent Intermolecular Interactions. J. Comput. Chem. 2016, 37 , 34–45. https://doi.org/10.1002/jcc.23946.
Grabowski, S. J.; Sokalski, W. A. Are Various Sigma-Hole Bonds Steered by the Same Mechanisms? ChemPhysChem 2017. https://doi.org/10.1002/cphc.201700224.
Wang, H.; Wang, W.; Jin, W. J. σ-Hole Bond vs π-Hole Bond: A Comparison Based on Halogen Bond. Chem. Rev. 2016, 116 (9), 5072–5104. https://doi.org/10.1021/acs.chemrev.5b00527.
Reviews:
Edwards, A. J.; Mackenzie, C.; Spackman, P.; Jayatilaka, D.; Spackman, M. A. FDHALO17: Intermolecular Interactions in Molecular Crystals: What’s in a Name? Faraday Discuss. 2017. https://doi.org/10.1039/C7FD00072C.
Costa, P. J. The Halogen Bond: Nature and Applications. Physical Sciences Reviews 2017, 2(11). https://doi.org/10.1515/psr-2017-0136.
Kozuch, S.; Martin, J. M. L. Halogen Bonds: Benchmarks and Theoretical Analysis. J. Chem. Theory Comput. 2013, 9 (4), 1918–1931. https://doi.org/10.1021/ct301064t.
Astatine:
Guo, N.; Maurice, R.; Teze, D.; Graton, J.; Champion, J.; Montavon, G.; Galland, N. Experimental and Computational Evidence of Halogen Bonds Involving Astatine. Nature Chemistry 2018, 10(4), 428–434. https://doi.org/10.1038/s41557-018-0011-1.
Receptors:
Molina, P.; Zapata, F.; Caballero, A. Anion Recognition Strategies Based on Combined Noncovalent Interactions. Chem. Rev. 2017. https://doi.org/10.1021/acs.chemrev.6b00814.
Scheiner, S. FDHALO17: Comparison of Halide Receptors Based on H, Halogen, Chalcogen, Pnicogen, and Tetrel Bonds. Faraday Discuss. 2017. https://doi.org/10.1039/C7FD00043J.
Reviewer 2 Report
It is very comprehensive review on various non-covalent interactions. I have the feeling, however, that polar dihydrogen interactions DHB (e.g. BH--HN) and the related non-polar DHB (e.g. intra- and inter-molecular CH---HC) shall be also referenced. The latter intra-molecular CH--HC, though quite controversial (e.g. cis-2-butene, biphenyl), shall be also pointed out to stimulate a scientific debate in order to fully unveil the true nature of intra-molecular "steric-crowding".
Author Response
First of all, we would like to thank this reviewer for his/her careful reading of the manuscript, suggestions, corrections. Our answers and corrections follow:
Comment:
It is very comprehensive review on various non-covalent interactions. I have the feeling, however, that polar dihydrogen interactions DHB (e.g. BH--HN) and the related non-polar DHB (e.g. intra- and inter-molecular CH---HC) shall be also referenced. The latter intra-molecular CH--HC, though quite controversial (e.g. cis-2-butene, biphenyl), shall be also pointed out to stimulate a scientific debate in order to fully unveil the true nature of intra-molecular "steric-crowding".
Answer:
Reviewer 2 should take into account that hydrogen bonds have been excluded from our review; consequently, DHBs were not discussed.
Round 2
Reviewer 1 Report
I do not want to hold the publication of this article. As said before, I believe it is a good review, I just do not agree with the extrapolation of so many bonds to the schema of hole interactions.
The definition of non-covalent bonds of page 3 is still quite incorrect, considering (as I said before) that even London forces are NCIs (this comment was not tackled by the authors).
The authors say that they do not want to include cations to alkali bonds, but they still do!
In fig. 7, change “ZnCl2” to the correct name. Still, I am unsure that it is a real “Hg-bond”, since the Hg-Cl-Hg angle should be close to 90 degrees. If it is close to 180, then there is a chance that the Hg acts as a Lewis base, and the Cl the Lewis acid. I really do not know how this bond is, but it is suspicious.
And yes, I would prefer if the authors remove the Au-N interaction, as it is a clear DCD coordination.
In summary, there is a difference of opinion between the authors and this reviewer. But opinions are not facts, so I will not give a judgement on this matter.